# Notch Signaling in Breast Cancer: A Role in Drug Resistance

**DOI:** 10.3390/cells9102204

**Published:** 2020-09-29

**Authors:** McKenna BeLow, Clodia Osipo

**Affiliations:** 1Integrated Cell Biology Program, Loyola University Chicago, Maywood, IL 60513, USA; mbelow@luc.edu; 2Department of Cancer Biology, Loyola University Chicago, Maywood, IL 60513, USA; 3Department of Microbiology and Immunology, Loyola University Chicago, Maywood, IL 60513, USA

**Keywords:** Notch, breast cancer, resistance, cancer stem cells

## Abstract

Breast cancer is a heterogeneous disease that can be subdivided into unique molecular subtypes based on protein expression of the Estrogen Receptor, Progesterone Receptor, and/or the Human Epidermal Growth Factor Receptor 2. Therapeutic approaches are designed to inhibit these overexpressed receptors either by endocrine therapy, targeted therapies, or combinations with cytotoxic chemotherapy. However, a significant percentage of breast cancers are inherently resistant or acquire resistance to therapies, and mechanisms that promote resistance remain poorly understood. Notch signaling is an evolutionarily conserved signaling pathway that regulates cell fate, including survival and self-renewal of stem cells, proliferation, or differentiation. Deregulation of Notch signaling promotes resistance to targeted or cytotoxic therapies by enriching of a small population of resistant cells, referred to as breast cancer stem cells, within the bulk tumor; enhancing stem-like features during the process of de-differentiation of tumor cells; or promoting epithelial to mesenchymal transition. Preclinical studies have shown that targeting the Notch pathway can prevent or reverse resistance through reduction or elimination of breast cancer stem cells. However, Notch inhibitors have yet to be clinically approved for the treatment of breast cancer, mainly due to dose-limiting gastrointestinal toxicity. In this review, we discuss potential mechanisms of Notch-mediated resistance in breast cancer cells and breast cancer stem cells, and various methods of targeting Notch through γ-secretase inhibitors, Notch signaling biologics, or transcriptional inhibitors. We also discuss future plans for identification of novel Notch-targeted therapies, in order to reduce toxicity and improve outcomes for women with resistant breast cancer.

## 1. Breast Cancer and Drug Resistance

Breast cancer remains the second leading cause of cancer-related death in women worldwide. It is predicted that approximately 42,170 women will die of this disease in 2020 [1]. Breast cancer is a heterogeneous disease often classified into diverse molecular subtypes as defined by protein expression of the Estrogen Receptor (ER), Progesterone Receptor (PR), and/or the Human Epidermal Growth Factor Receptor 2 (HER2) [2]. Tumors that are classified as ER+ and/or PR+ overexpress the ER, PR, or both proteins and can be effectively targeted using endocrine therapy. Endocrine therapy includes the use of a selective estrogen receptor modifier, (SERM), an aromatase inhibitor (AI), and/or a selective estrogen receptor degrader (SERD). Aromatase inhibitors have been shown to significantly improve disease-free survival (DFS) after five years of treatment for post-menopausal women [3,4]. Tamoxifen, a SERM, is approved for pre-menopausal women and has been shown to significantly increase overall survival after five years of adjuvant therapy [5,6,7]. Alternatively, HER2+ tumors overexpress the receptor tyrosine kinase, HER2, due to a gene amplification of the proto-oncogene, *ERBB2*. These tumors are effectively targeted by using first-line humanized, monoclonal antibodies, including trastuzumab and pertuzumab [8,9,10], or, later, second-line tyrosine kinase inhibitors, such as lapatinib [11] or neratinib [12,13,14]. Breast cancers that lack expression of the ER, PR, and overexpression of HER2 are classified as triple negative and treated with cytotoxic chemotherapy due to lack of approved targeted therapy [15,16,17,18].

Though targeted therapies are efficacious in non-advanced disease, a reported 20–50% of HER2+ metastatic tumors have an inherent resistance to trastuzumab or other HER2-targeted agents, and 10–15% will acquire resistance within the first year of treatment [10,19,20]. Similarly, 30% of ER+ tumors display intrinsic resistance to one or more forms of endocrine therapy, and approximately 30% acquire resistance to tamoxifen [21]. Triple negative breast cancers are inherently resistant to HER2 or ER-targeted therapies and thus are treated with cytotoxic chemotherapy, such as taxane-based, platin-based, and other DNA-damaging agents [22]. Therefore, it is critical to elucidate potential mechanisms of drug resistance in breast cancer, in order to improve survival outcomes for women.

## 2. Overview of the Notch Signaling Pathway

Notch signaling is an evolutionary conserved pathway, originally discovered through investigations of *Drosophila* wing development [23] and has since grown into an increasingly large field of study for cancer biologists. This intricate pathway mediates normal stem cell differentiation, cell fate, and organ development [24,25]. However, its dysregulation and role in promoting cellular transformation has led to further investigations of the role of Notch in a variety of cancers [26].

There exist four known mammalian Notch receptors, Notch1, Notch2, Notch3, and Notch4. Each receptor is translated as a single polypeptide that is subsequently cleaved in the Golgi-apparatus by a furin-like convertase. The resulting cleaved protein is delivered to the plasma membrane as a heterodimeric protein containing an extracellular domain tethered to the transmembrane and intracellular domains by a calcium cation (Figure 1). Upon interaction of the extracellular domain with one of its ligands that include Jagged-1 (JAG1), Jagged-2 (JAG2), Delta-like 1 (DLL1), Delta-like 3 (DLL3), or Delta-like 4 (DLL4), through cell-to-cell contact (Figure 1 and Figure 2), the extracellular portion of the receptor is pulled away from the transmembrane/intracellular domains by ligand-mediated endocytosis. The remaining transmembrane portion of the receptor (Notch^TM^) is first cleaved by a disintegrin and metalloprotease (ADAM17 or ADAM10), resulting in a product: Notch extracellular truncation (NEXT). NEXT is subsequently cleaved by the γ-secretase complex releasing the intracellular portion of Notch (Notch^IC^). Notch^IC^ is translocated from the cytoplasm to the nucleus where it binds to the CSL (CBF-1/RBPJ-κ in *Homo sapiens/Mus musculus*, respectively, Suppressor of Hairless in *Drosophila melanogaster*, Lag-1 in *Caenorhabditis elegans*) transcription factor. The interaction of Notch^IC^ with CSL replaces corepressors with coactivators including the transcriptional activator Mastermind1 (MAML1) at regulatory sequences of gene targets (Figure 2). This allows for transcriptional activation of Notch target genes [27,28].

Some of the earliest known targets of Notch signaling include transcriptional repressors, such as the hairy/enhancer of split (*HES*) genes, as well as the HES subfamily members *HEY1*, *HEY2*, and *HEYL* [29,30]. These *HES/HEY* genes are critical cell-fate regulators during development and tissue renewal. In addition to this, cell-cycle regulators such as c-Myc [31] and cyclin D1 [32] are directly activated by Notch signaling. Dysregulation of Notch signaling, such as activating Notch receptor mutations, overexpression of ligands and/or receptors, and/or overexpression of its target genes, contributes to increased proliferation, cell transformation, and increased drug resistance in cancers of the breast, multiple myeloma, prostate, T-cell acute lymphoblastic leukemia, and others [33].

## 3. A Role for Notch in Breast Cancer

### 3.1. Notch as a Breast Oncogene

It has been shown that Notch is an oncogene in the breast, as overexpression of Notch1^IC^ [34,35], Notch3^IC^ [35], or Notch4^IC^ [36,37] is sufficient for transformation of normal breast epithelial cells into cancer cells. Overexpression of Notch1 and/or Jagged1 predicts the poorest overall survival outcome for women with breast cancer [38,39]. Early studies show that normal breast tissue has high expression of the negative Notch regulator, Numb, and that its expression is lost in breast tumors [40]. Treatment with the proteasome inhibitor MG-132 led to increased Numb expression in primary cultures of human breast tumor cells and decreased Notch transcriptional activity. Based on these findings, Stylianou and colleagues investigated whether Notch was aberrantly activated in breast cancer and how this may impact cellular transformation. Upon stable overexpression of Notch1^IC^ in the non-transformed breast cell line MCF-10A, they were able to demonstrate cellular transformation via changes in cell shape, increased cell growth, colony formation, and resistance to apoptosis. Importantly, overexpression of Numb in the ER+ breast cancer cell line MCF-7 resulted in decreased Notch^IC^ accumulation, inhibition of colony formation, and accumulation of E-cadherin, suggesting that transformation of these cells had been reversed [41]. Together, these data demonstrate that increased Notch activity and/or deregulation of Notch leads to the transformation of normal breast cells into cancer cells.

### 3.2. Notch as a Prognostic Biomarker

Expression and activation of Notch in primary breast tumors has been used to assess if Notch signaling is a prognostic and/or predictive biomarker. For example, overexpression of Notch1 and Jagged1 predict the poorest overall outcome for women with breast cancer, with a predicted mortality of 63% in women with JAG1^high^-expressing tumors, compared to 32% in JAG1^low^-expressing tumors. Furthermore, women with Notch1^high^-expressing tumors had a 66% mortality rate, compared to 30.5% for Notch1^low^-expressing tumors [39]. A study conducted by Yao and colleagues (2011) identified that expression of Notch1 and Notch4 proteins was cytoplasmic in ER+ breast tumors, compared to ER− tumors. In conjunction with this, Ki67 expression, a nuclear protein associated with proliferation [42], significantly correlated with Notch1 nuclear expression and Notch4 membrane and cytoplasmic expression in ER+ tumors. Further research demonstrated that Notch1 and Notch4 immunoreactivity significantly correlated with tumor grade and Ki67 expression in triple-negative breast tumors [43]. These findings and others [38,44,45,46,47,48,49] provided support for Notch as a poor prognostic biomarker in breast cancer.

### 3.3. Notch and Epithelial-to-Mesenchymal Transition

Additionally, Notch signaling promotes activation of genes required for epithelial-to-mesenchymal transition (EMT). The EMT process is a prerequisite for metastatic spread of the tumor from the primary site (breast) to distant organs (i.e., lymph nodes, bone, brain, lung, or liver). Data suggest that positive regulation of *SLUG* by Jagged1-mediated activation of Notch^IC^ results in the repression of E-cadherin, thus allowing for EMT in breast cancer cells [50,51,52]. Jagged1 overexpression alone is associated with increased bone metastasis, and disruption of the Notch pathway by using the γ-secretase inhibitor (GSI) MRK-003 reversed Jagged-1-mediated bone metastasis in mice [53,54]. Hypoxia-induced Notch activation promotes EMT in breast cancer cells, as shown by increased expression of Notch targets *HEY2* and *HES1*, downregulation of E-cadherin and β-catenin, and increased cell migration and invasion of breast cancer cells cultured in low-oxygen conditions [55,56].

Importantly, EMT gives rise to a cancer stem-cell-like phenotype in immortalized human mammary epithelial cells (HMECs) [57,58,59]. Studies show that HMECs-overexpressing SNAIL or TWIST alone successfully undergo EMT, and most if not all transitioned cells were CD44^high^/CD24^low^ [58], which are markers of breast cancer stem cells. Furthermore, BCSCs have cell fate plasticity and can change from epithelial to mesenchymal phenotypes [60,61,62]. Metastatic breast cancer has an “incurable” nature, with an estimated five-year survival of only 26% [63], and it has been proposed that cancer stem cells are responsible not only for breast metastases, but also resistance to therapy and disease recurrence [64,65,66].

## 4. Notch and Breast Cancer Stem Cells (BCSCs)

### 4.1. Stem Cell Markers

One proposed theory for drug resistance is that a small population of cells referred to as breast cancer stem-like cells (BCSCs) within the bulk primary tumor are inherently resistant to many forms of targeted or cytotoxic therapy. These BCSCs survive therapy and remain dormant until they are reactivated to proliferate, depending on the microenvironment (Figure 3). A small population of CD44^+^/CD24^−/low^ cells were originally isolated from patient tumors in 2003 by Al-Hajj et al., in which these cells were found to have high tumor-initiating potential, 10- to 50-fold greater than CD44^+^/CD24^+^ cells, when injected into immune-compromised mice. These cells were coined “cancer stem cells” due to their high tumor initiating potential and their ability to form distinct populations of stem-like and differentiated cells within the bulk tumor [67].

The use of stem cell markers has allowed for the study of distinct stem cell populations within the bulk population of breast cancer cell lines. A study using the inflammatory breast cancer cell line SUM149, found that ALDEFLOUR+ cells comprised approximately 5% of the total cell population, and of this ALDEFLOUR+ population, 13% of cells were CD44^+^/CD24^−^, compared to only 3% in ALDEFLOUR- cells. However, CD44^+^/CD24^−/low^ cells were not predictive of metastasis or patient survival, whereas ALDH1 or Jagged1 expression correlated with development of metastasis and decreased survival of patients with inflammatory breast cancer [68,69]. ALDH is expressed in normal tissue and is responsible for the oxidation of intracellular aldehydes [70]. However, its elevated activity in neural and hematopoietic stem cells [71,72,73,74], along with multiple myeloma and acute myeloid leukemia cells [75,76], led researchers to investigate its role in mammary stem cells. Ginestier and colleagues (2007) established that high ALDH activity was found in normal breast epithelial along with cells collected from human-derived breast cancer xenografts. Using the ALDEFLOUR assay, which utilizes a fluorescent aldehyde to detect ALDH activity [77], they were able to establish that ALDEFLOUR positive cells had high tumorigenicity, and ultimately that the isoform ALDH1 was a viable diagnostic biomarker and predictor of poor clinical outcomes [77,78]. These data suggest that there are distinct potentially overlapping populations of BCSCs and warrant further investigations into the mechanistic differences between them.

It has recently been shown that there are distinct types of cancer stem cells. For example, *TWIST1* promotes EMT through repression of E-cadherin, while simultaneously promoting a stem cell phenotype through the downregulation of CD24 [62,79,80]. Alternatively, mammalian Y-box binding protein 1 (YB1) both upregulates translation of *TWIST1* and *SNAI1*, along with stem cell markers p63 and CD44, while at the same time repressing translation of CD24 [62,81]. CD44^+^/CD24^−/low^ cells express mesenchymal properties, such as low E-cadherin levels, increased expression of Vimentin, and a quiescent phenotype, whereas ALDH^+^ cells express relatively high levels of E-cadherin with low levels of Vimentin and are generally more proliferative, which is considered to be an epithelial phenotype [60]. Current studies are investigating the plasticity of BCSCs to transition between EMT and mesenchymal to epithelial transition (MET) states and how this may contribute to therapy resistance and metastatic-tumor development [61,62,82,83].

### 4.2. Mammosphere Forming Efficiency

The mammary glands are derived from mammary stem cells that differentiate into luminal and myoepithelial progenitors, and these further differentiate into luminal and myoepithelial cells. Mammary stem cells were originally investigated by using a unique suspension cell culture technique [11]. Single cell suspensions were cultured in a mammosphere-forming medium, as the term “mammosphere” is derived from the ability of the cells to proliferate in suspension in the form of a sphere, as previously seen by using neuronal stem cells which formed “neurospheres” [84,85]. Interestingly, the use of the mammosphere formation assay has shed light on a role on Notch signaling as a key regulator of cell fate of normal mammary progenitor cells and allowed present studies to delve deeper into the role of Notch in breast cancer [86,87,88].

Dontu and colleagues were the first to use this assay to show activation of Notch signaling promotes proliferation and self-renewal of mammary stem/progenitor cells. Further, the investigators demonstrated that Notch signaling was also required for the lineage commitment of mammary progenitors to myoepithelial cells in vivo as either Notch4 blockade or a GSI inhibited the myoepithelial lineage commitment [86]. This technique was then utilized to isolate Ductal Carcinoma in Situ (DCIS) cells from human breast tissues, and researchers were able to demonstrate a direct role for Notch by inhibiting the mammosphere forming ability of these cells, using the GSI DAPT [88]. Interestingly, Notch target genes were shown to be elevated in mammospheres derived from various breast cancer cell lines [89]; and ALDEFLOUR+ breast cancer cells had increased mammosphere-forming efficiency (MFE), compared to ALDEFLOUR− cells, which correlated with an increased expression in Notch2 mRNA, a known promoter of mammary stem cell self-renewal [90].

### 4.3. Extreme Limiting Dilution Assay

In addition to an in vitro cell culture assay, a reliable and valid in vivo assay also exists to investigate the contribution of CSCs on tumor development. The Extreme Limiting Dilution Assay (ELDA) estimates the frequency of cancer stem cells within a given cell dilution. Serial dilutions of cancer cells (i.e., 10, 100, 1000, 10,000, 100,000, and 1,000,000) are injected into immune-compromised mice, to assess the rate of tumor development and tumor burden. Once the number of mice with tumors is determined, the frequency of cancer stem cells within the bulk population is estimated, using a web tool at the website http://bioinf.wehi.edu.au/software/elda/. This web tool generates an estimated 95% confidence interval for all datasets, including situations where there is 0% or 100% response at all dilutions, making this an optimal design for studying frequency of cancer stem cells within a bulk cell population [91].

ELDAs have been used extensively in investigating BCSCs, either alone or in conjunction with the mammosphere formation assay. The assays together have proven to be valuable tools investigating the role of Notch in BCSCs. Harrison et al. (2010) injected mice with limiting dilutions of MCF-7 BCSC-enriched, BCSCs-depleted populations, or a mixed population, and showed that the BCSC-enriched population formed tumors by using 1 × 10^4^ cells at a frequency of 75%, compared to BCSC-depleted cells. Meanwhile, the mixed population required 1 × 10^6^ cells to achieve 75% tumor frequency [92]. Additionally, MCF-7 cells stably expressing a dominant-negative mutant of MAML (dnMAML) injected into mice showed an approximately 60% decrease in CSC frequency, compared to MAML-GFP-expressing cells, as calculated by the ELDA web tool, following a 90-day period post-injection [93].

### 4.4. BCSCs and Drug Resistance

CSC studies are often focused on understanding the molecular mechanisms behind their inherent resistance to various therapies. The cancer stem-cell hypothesis proposes that tumors arise from dysregulated tissue-specific stem cells or their progeny, referred to as transient amplifying cells, which in turn give rise to key stem cell properties such as self-renewal, differentiation, and resistance to therapies. Early studies showed that CD44^+^/CD24^−^ cells were enriched in the bulk cell population and mammospheres were elevated following neoadjuvant chemotherapy in isolated tumor cells, regardless of receptor subtype [94]. Additionally, tumors treated with sequential paclitaxel and epirubicin-based chemotherapy showed an enrichment of ALDH^+^ tumor cells with no significant changes in CD44^+^/CD24^−^ populations [95], further suggesting distinct stem cell populations may exist within breast cancer tumors and may contribute to drug resistance.

Recent work has uncovered mechanisms suggesting a clear role for Notch1 and Notch4 in drug-resistant BCSCs. Studies have shown that exposure to chemotherapy, such as doxorubicin or docetaxel, as well as an anti-estrogen, such as tamoxifen or fulvestrant, results in an enrichment of ALDH^+^ BCSCs that are resistant to these therapies [66,96]. Additionally, upon knockdown of Notch1 via short-interfering RNA (siRNA), or inhibition of Notch1 via the bioactive compound Psoralidin, ALDH^+^ cells were growth inhibited, formed fewer mammospheres, had increased apoptosis, and limited tumor growth in mice. Inhibition of Notch4 by the GSI RO4929097 in vivo resulted in decreased mammospheres of tumor-isolated cells and severely restricted tumor-initiating cell frequency, using ELDA [96]. More recently, Shah et al. (2018) demonstrated that HER2+ breast cancer cells treated with lapatinib were enriched for high membrane-Jagged1-expressing BCSCs. These Jagged1^high^-expressing cells were inherently resistant to lapatinib and possessed high mammosphere forming efficiency and tumor initiating potential [97]. Furthermore, Baker et al. (2018) showed that the mechanism for trastuzumab resistance, high mammosphere forming, and tumor initiating potential was Notch1-mediated repression of PTEN in HER2+ breast cancer cells [98]. Although these studies offer clear examples that Notch signaling contributes to resistance and increased BCSC survival, further work is necessary to elucidate other regulators and mechanisms of Notch-mediated resistance in BCSCs.

## 5. Notch-Mediated Resistance to Breast Cancer Treatment

### 5.1. Estrogen Regulation of Notch Signaling

Women with ER+ breast cancer are treated with endocrine therapy, such as tamoxifen for pre-menopausal women, an aromatase inhibitor for post-menopausal women, or fulvestrant as second-line therapy [99]. Tamoxifen or fulvestrant are competitive inhibitors of 17β-estradiol by directly binding to the ligand binding domain of the ER. In this manner, tamoxifen or fulvestrant inhibit 17β-estradiol-mediated ER signaling in the cancer cell [100]. An aromatase inhibitor indirectly targets ER signaling by decreasing synthesis of 17β-estradiol from local androgens mediated by the aromatase enzyme (Figure 4) [101]. This is a very effective strategy to decrease proliferation of ER+ breast cancer cells and reduces bulk tumor size [102,103,104]. Unfortunately, 50–60% of all early breast cancer cases, and nearly all advanced cases, develop resistance to one form of endocrine therapy [105,106]. During the last several decades, Notch signaling has emerged as a mechanism by which ER+ breast cancer cells develop resistance to endocrine therapy [96,107,108].

Studies have shown that ER+ breast cancer cell lines treated with estrogen-deprivation therapy, 4-hydroxytamoxifen, or fulvestrant have increased expression of the cleaved forms of Notch1 (Notch1^IC^) and Notch4 (Notch4^IC^) [107]. Specifically, endocrine-therapy-induced Notch1^IC^ and Notch4^IC^ increased CSL-driven reporter activity in ER+ MCF-7 and T47D cells, suggesting that, while 17β-estradiol-mediated activation of ER maintained low Notch activity, antagonizing ER increased Notch signaling (Figure 4).

It has been established that mutations in the hormone binding domain of the ER, specifically the Y537S mutation, result in a constitutively active ER and confer endocrine therapy resistance in ER+ breast cancer [109]. Recent studies show that Y537S-*ESR1*-expressing MCF-7 cells were enriched for CD44^+^/CD24^−^ cell populations with increased mammosphere-forming efficiency, compared to wild-type *ESR1* expressing cells, as well as elevated mRNA levels of all four Notch receptors and enhanced protein expression of Notch4, Notch4^IC^, Jagged1, DLL1, and DLL3 [110]. Importantly, treatment with the GSI, DAPT or RO4929097 significantly decreased the mammosphere forming efficiency of the mutant expressing cells, suggesting that their BCSC activity is Notch dependent. As Notch signaling has been shown to be oncogenic in the breast, these data indicated that one consequence of endocrine therapy is activation of Notch to promote survival of BCSCs and thus promote resistance to endocrine therapy.

### 5.2. Crosstalk of Notch and HER2 Signaling

HER2+ breast cancers overexpress the HER2 receptor due to a gene amplification of the *ERBB2* gene. HER2 is the second member of the HER family of receptor tyrosine kinases which include HER1 (EGFR), HER3, and HER4. When HER2 is overexpressed, it is the preferred dimer partner for the other receptors, depending on expression and growth factor availability. Dimerization can be heterotypic or homotypic, depending on expression patterns (Figure 5). HER2 homodimerization does not require growth factor, while heterodimerization of HER2 with other family members requires growth factor. Upon dimerization, the kinase domain is activated, resulting in cis- and trans-phosphorylation of several tyrosine residues. The phospho-tyrosines recruit adaptor proteins that activate downstream kinase cascades, including the Ras-Raf-MEK-ERK, PKC, PI-3K, and JAK-STAT pathways (Figure 5). These signaling cascades result in transcription-factor-mediated expression of genes required for cell growth, migration, and invasive (Figure 5). HER2 and the activated pathways are targeted by treatments that include humanized, monoclonal antibodies trastuzumab and pertuzumab in combination with a taxane-based chemotherapy [111]. Once tumors progress on these first-line therapies, second line therapies are given in the form of HER tyrosine kinase inhibitors, lapatinib [11] or neratinib [12], or antibody conjugates, including T-DM1 (ado-trastuzumab emtansine) [112].

HER2 antibodies have been shown to reduce expression of HER2 receptors on the cell surface, prevent heterodimerization, mediate antibody-directed cell cytotoxicity (ADCC), and inhibit proliferation of HER2+ breast cancer cells [113]. However, tyrosine kinase inhibitors competitively bind ATP-binding sites within the kinase domain of the receptor to inhibit auto- and trans-phosphorylation, leading to kinase inactivity and inhibition of downstream signaling [114]. Though these therapies are highly efficient in reducing tumor size and improving survival, HER2-driven cancers are known to be aggressive, with 40–50% of HER2+ patients developing brain metastases [115], and 70% of patients present with intrinsic or secondary resistance to targeted therapies [116]. Mechanistic studies showed resistance could be due direct regulation of the *ERBB2* gene by Notch1 [117], and in turn increased HER2 may activate Notch [118], possibly creating a positive feedback loop. Others have demonstrated that inhibition of HER2 results in an upregulation of Notch activity, suggesting HER2 may negatively regulate Notch signaling [119]. Specifically, HER2 blockade by trastuzumab or a tyrosine kinase inhibitor increased *NOTCH1* and *JAG1* expression, activating Notch1 signaling contributing to resistance [120]. Trastuzumab-resistant cells have increased expression of *NOTCH1*, *JAG1*, and their targets, including *HEY1*, *DTX1*, and *HES5*, and downregulation of Notch1 via siRNA sensitized these cells to trastuzumab treatment (Figure 5) [120]. Interestingly, blocking Notch activity with a GSI results in downregulation of HER2 expression on the cell surface, as well as at the mRNA level in HER2+ breast-cancer-derived mammospheres [121], and research shows promising results of combination therapy targeting Notch and HER2 effectively targets ductal carcinoma in situ (DCIS) stem cells [119] or prevents recurrence in pre-clinical HER2+ tumors [122]. Thus, elucidating pathway crosstalk between HER2 and Notch signaling has proven to be essential in furthering our understanding of drug resistance and cancer recurrence, and must be continued in order to fully uncover mechanisms and develop novel therapeutics for the clinic.

## 6. Notch Inhibition as a Therapeutic Approach

### 6.1. γ-Secretase Inhibitors

In vitro studies using ER+ breast cancer lines provided the initial evidence that Notch activation contributes to resistance to endocrine therapy and that targeting Notch1 or Notch4 by using genetic or pharmacologic means reversed this resistance. One of the first Notch inhibitors that was designed was a γ-secretase inhibitor (GSI). The GSI was shown to inhibit the final cleavage step of Notch, thus preventing the release of Notch^IC^ and inhibiting activation of Notch target genes (Figure 2). For example, Rizzo et al. (2008) demonstrated that inhibition of the γ-secretase complex using the GSI, Z-Leu-Leu-Nle-CHO abrogated enhanced Notch activity in response to 4-hydroxytamoxifen, estrogen deprivation, or fulvestrant. The use of the GSI led to strong inhibitory effects on the growth of ER+ T47D cells. They also showed that, when combined with tamoxifen, T47D-A18 xenograft tumors treated with the GSI significantly regressed and experienced extensive cell death in vivo [107]. As mentioned previously, Simões et al. (2015) found that cells isolated from tumors treated with RO4929097 GSI in vivo had significantly reduced tumor-initiating potential 90 days post-implantation in mice and reduced ALDH activity [96]. One study comparing effects of Z-Leu-Leu-Nle-CHO, MRK-003, and LY-411,575 GSIs found a significant reduction in mammosphere-forming efficiency following treatment with Leu-Nle-CHO or MRK003, with a non-significant reduction seen in cells treated with LY-411,575 [89]. Additionally, it has been established that MRK-003 effectively induces apoptosis in mammosphere-derived stem-like cells both in vitro and in vivo, suggesting a promising approach using a GSI to target the BCSC population [89,123].

Differential effects of GSIs on breast tumor growth could be due to their differences in chemical structures. Some GSIs are transition state analogs (i.e., Z-Leu-Leu-Nle-CHO, LY-685458, YO-01027, and RO-4929097), while others are non-transition analogs (i.e., Dapt, Compound E, MRK-003, MK-0572, and PF-03084014). Of these GSIs, MK-0752, RO-4929097, and PF-03084014 have been tested in phase I and/or II clinical trials, in both early and advanced disease [124]. However, to date, GSIs have yet to be approved for treatment of breast cancer. This is thought to be due to lack of antitumor efficacy and, in some cases, severe gastrointestinal toxicity [125].

### 6.2. Notch Signaling Biologics

In addition to targeting the γ-secretase complex, Notch receptors and their ligands can be targeted directly, using monoclonal antibodies. Monoclonal antibodies prevent Notch receptor interactions with its ligands (Figure 2). This strategy offers a more selective method to target Notch signaling in tumors and avoid unwanted toxicity associated with GSIs. The monoclonal antibody OMP-59R5 was developed to selectively target Notch2 and Notch3 receptors and has shown stable disease efficacy in clinical trials of triple negative breast cancer [125]. Another study showed an antibody targeted against the EGF repeats in the ligand-binding domain of Notch1 significantly inhibited cell proliferation of MCF-7 and MDA-MB-231 cells. Importantly, it specifically targeted the BCSC population by modulating stem cell gene expression and inducing apoptosis [126].

### 6.3. Notch Transcriptional Inhibitors

Notch receptors are transcriptional activators, and the development of strategies to inhibit transcription has been challenging. One class of agents known as “stapled peptides” were developed to block the interaction between MAML1 and the Notch^IC^ to prevent transcriptional activity (Figure 2) [127]. Moellering and colleagues (2009) were the first to demonstrate that a hydrocarbon-stapled peptide derived from a dominant negative form MAML1 known as SAHM1 competitively bound Notch1-CSL transcriptional complex to repress Notch1 target gene expression in T-cell acute lymphoblastic leukemia (T-ALL) [127]. In addition to MAML1, other transcription complex partners including CSL have been investigated as potential targets to inhibit Notch signaling. For example, a small molecule inhibitor of the Notch-CSL-MAML1 ternary complex has shown antitumor efficacy in breast cancer and other cancer models [128]. Other direct targeting approaches have included inhibiting CSL expression or activity resulting in decreased proliferation and growth of breast cancer cell lines [129,130,131]. These data suggest that targeting downstream signaling components of the Notch signaling pathway, including its transcriptional partners, may prove to be a more specific and possibly efficacious strategy in treating Notch-driven breast cancers.

## 7. Future Ideas

Though a significant amount of work has been done to elucidate mechanisms of Notch-mediated resistance in breast cancer, future work will need to focus on development of selective and safe therapeutic strategies to inhibit this signaling pathway. A promising approach is identification Notch downstream targets that are specifically expressed in breast tumors as biomarkers for prognosis or predictive markers of sensitivity to treatment. The identification of breast tumor specific Notch targets was conducted by using a pre-surgical window trial in women with early stage ER+ breast cancer [132]. In this trial, 22 women received neo-adjuvant letrozole (an aromatase inhibitor) or tamoxifen for 14 days, followed by addition of MK-0572 GSI for 11 days. Microarray analysis was conducted to measure RNA transcript expression before GSI and after GSI treatment. Many canonical Notch gene transcripts were decreased in response to GSI treatment. Furthermore, many novel Notch target transcripts were also regulated by the GSI. One of these was death-domain-associated protein 6 (DAXX), which was upregulated with GSI treatment [132]. Specifically, *DAXX* was identified to be a novel Notch1 target gene, and high expression of Notch1 repressed *DAXX* (unpublished data). Recently, Peiffer et al. (2019) demonstrated that high DAXX expression mediated by ER activation potently inhibited BCSCs in vitro and in vivo [133]. Upon endocrine therapy, DAXX was depleted due to protein instability and this decrease in DAXX increased *NOTCH4*, pluripotent stem cell genes (*SOX2*, *OCT4*, and *NANOG*), and BCSC phenotypes. From these data, a novel therapeutic approach was used to stabilize the DAXX protein, using a phytoestrogen in combination with estrogen deprivation therapy to potently inhibit BCSCs and delay tumor development [134]. In addition to pursuing downstream targets of Notch, investigating upstream regulators of Notch could also lead to promising therapeutic tools. For example, Polyomavirus enhancer activator 3 (PEA3), a member of the ETS transcription factor family, is overexpressed in triple negative breast cancer and is associated with metastasis [135]. Clementz and colleagues (2011) demonstrated *NOTCH1* and *NOTCH4* are transcriptionally activated by PEA3 in breast cancer cell lines, and knockdown of PEA3 or treatment with a GSI resulted in a G_1_ cell cycle arrest, increased apoptosis, and either treatment reduced tumor burden in mice [136]. Therefore, novel approaches either upstream or downstream of Notch signaling could provide new and more selective strategies to inhibit Notch, prevent survival of BCSCs, resistance to therapies, and ultimately prevent tumor recurrence.

## 8. Concluding Thoughts

The Notch signaling pathway is an evolutionary conserved pathway that regulates key developmental processes and stem cell survival. It is proposed that cancer stem-like cells drive drug resistance in breast cancers, and targeting Notch activity has proven to be an effective approach to limiting this cell population in breast cancer. In vitro studies have shown Notch activity to be elevated in both bulk and BCSC populations in ER+ and HER2+ breast cancer cell lines, and this is enhanced upon treatment with therapies such as tamoxifen or trastuzumab, respectively. Upon knockdown of Notch, these cells undergo a significant growth arrest and lose their stem-like qualities, such as self-renewal, resistance, epithelial to mesenchymal transition (EMT), and tumor recurrence (Figure 3). Clinical studies investigating γ-secretase inhibitors have shown promising molecular changes in breast tumors when treated in combination with tamoxifen or letrozole. Other forms of Notch inhibition, such as monoclonal antibodies, targeting downstream targets, or post-translational modifiers have proven to be efficient and effective tools for breast cancer treatments both in vitro and *in vivo*. Future studies will continue to focus on development of effective and safe Notch pathway inhibitors as they move into preclinical and clinical trials, to improve survival outcomes for women with breast cancer.

## Figures and Tables

**Figure 1 cells-09-02204-f001:**
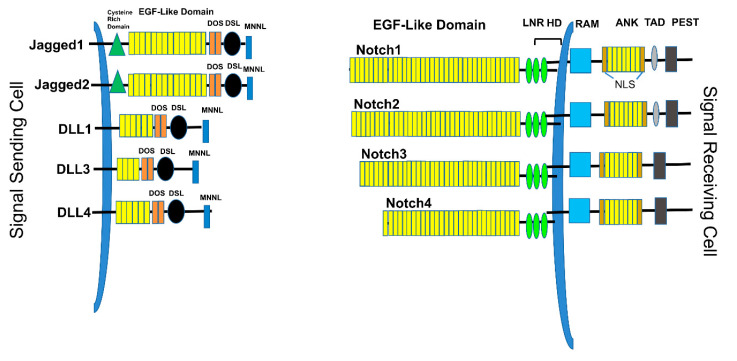
Notch ligands and receptors. The Signal-sending cell expresses Notch ligands, type I transmembrane proteins. These include Jagged1, Jagged2, DLL1, DLL3, and DLL4. Each ligand contains a short cytoplasmic tail, followed by the transmembrane domain, EGF-like repeats, a DOS domain, DLS binding domain, and MNNL domain. The signal-receiving cells express the Notch receptors, including Notch1, Notch2, Notch3, and Notch4. Each receptor contains EGF-like domains, LNR (Lin-Notch repeats), HD (heterodimerization domain), transmembrane domain, RAM (RBPJ-associated molecule), ANK (Ankyrin repeats), TAD (transactivation domain), and PEST (Pro-Glu-Ser-Thr) domains.

**Figure 2 cells-09-02204-f002:**
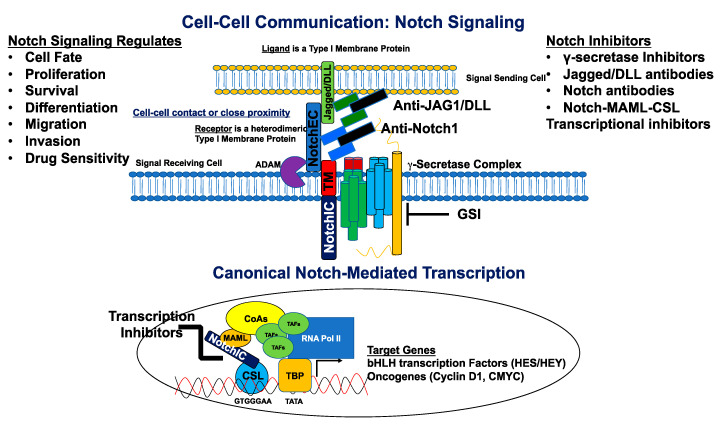
Canonical Notch signaling and inhibitors. The signal-sending cell expressing a Notch ligand (Jagged/DLL) is in close proximity to the signal-receiving cell expressing a Notch receptor. The ligand binds the Notch receptor through EGF-like repeats, and this interaction dislodges the extracellular portion of Notch (NotchEC) from the transmembrane portion (NotchIC-TM). NotchEC is endocytosed with the ligand into the signal sending cell. The removal of NotchEC exposes the ADAM cleavage site on the ecto-domain of the NotchIC-TM. ADAM-mediated cleavage creates a substrate (NEXT: Notch extracellular truncation) for the γ-secretase complex. This subsequent cleavage produces a product (NotchIC) that translocates to the nucleus. NotchIC binds to CSL and recruits MAML. MAML recruits other transcriptional co-activators and the RNA POL II initiation machinery to initiate transcription of Notch target genes including basic helix loop helix (bHLH) proteins of the HES and HEY family of transcriptional repressors. Other genes include the oncogenes, *CMYC* and *CCND1* (Cyclin D1) for initiation of the cell cycle. Notch signaling is inhibited by a variety of molecules that include γ-secretase inhibitors (GSIs), antibodies directed against Notch ligands and receptors, and transcriptional inhibitors that target the NotchIC-MAML-CSL ternary complex. Notch regulates cell fate, proliferation, survival, differentiation, migration, invasion, and sensitivity to cancer drugs.

**Figure 3 cells-09-02204-f003:**
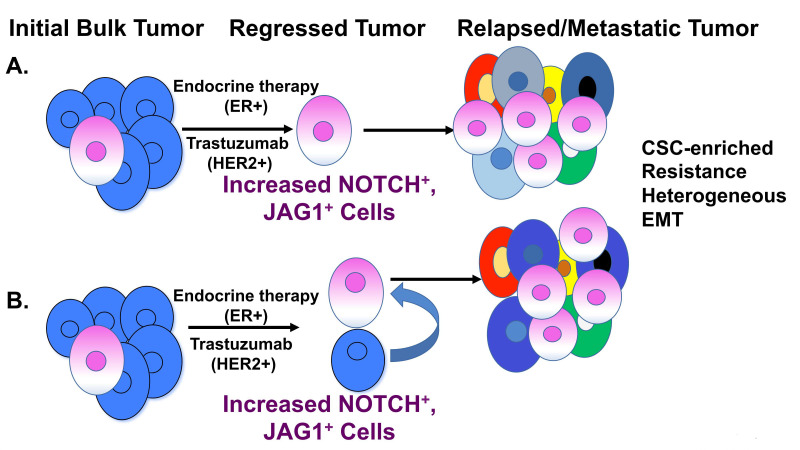
Role of breast cancer stem cells. (**A**) The initial cancer stem-cell hypothesis proposes that a small population of cells exist within the bulk tumor and these cells have inherent stem-like properties of drug resistance, self-renewal, and tumor-initiating potential. Upon treatment with endocrine or HER2-targeted therapy, such as trastuzumab, the majority of the bulk tumor regresses, leaving a small population of cells. These cells have high Notch activity due to increased Notch and/or Jagged1 (JAG1) expression. The Notch-enriched cells are cancer stem cells (CSC) with properties of drug resistance and give rise to a heterogeneous tumor that has undergone epithelial-to-mesenchymal transition (EMT). These CSCs are responsible for tumor relapse and metastatic spread. (**B**) An alternative hypothesis proposes that, upon regression of the bulk tumor, Notch signaling is elevated in both CSCs and non-CSCs. The Notch/Jagged-high-expressing CSCs have properties of self-renewal, drug resistance, and expand into a heterogeneous tumor with more aggressive features. Additionally, non-CSCs may de-differentiate to CSCs under prolonged treatment conditions due to increased Notch and/or Jagged expression. The converted CSCs may give rise to a heterogeneous tumor with drug resistance and metastatic properties.

**Figure 4 cells-09-02204-f004:**
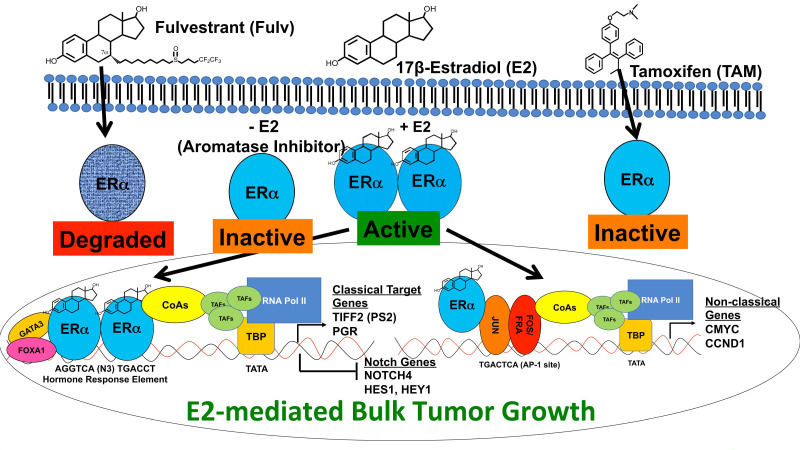
Estrogen signaling and regulation of Notch. The ERα ligand 17β-estradiol (E2) crosses the plasma membrane and binds with high affinity to the ERα monomer. Upon binding its ligand, ERα dimerizes and is in active conformation to enter the nucleus and activate transcription. Classical ER signaling requires the homodimers to bind hormone response elements. These hormone-response elements are frequently near FOXA1 and GATA3 binding elements. It is thought that ER homodimers are recruited to hormone-response elements through interaction with FOXA1 and/or GATA3. Once bound to the hormone-response elements, ER dimers recruit co-activator proteins, including p300 and other epigenetic modifying proteins, to unwind DNA and bring in the transcription initiation complex. Classical ER target genes, *PGR*, and *TIFF2* are expressed, while other genes, such as *NOTCH4, HES1*, and *HEY1*, are repressed. Non-classical ER signaling requires interaction of an ER monomer or dimer with other transcription factors (i.e., AP-1) to promote activation of AP-1-driven genes including *CMYC* and *CCND1* (Cyclin D1). In this fashion, E2 promotes tumor growth. Endocrine therapy includes treatment with estrogen-depleting (-E2) agents, such as aromatase inhibitors, and direct competitive inhibition, using tamoxifen (TAM) or fulvestrant (Fulv). TAM is an inhibitor of ER activity in breast cancer cells. Fulv is an ER degrader, and this results in complete downregulation of ER and ER signaling. Targeting ER expression and activity inhibits tumor growth. However, the consequence of ER inhibition is Notch activation and subsequent enrichment of Notch-high CSCs and resistance.

**Figure 5 cells-09-02204-f005:**
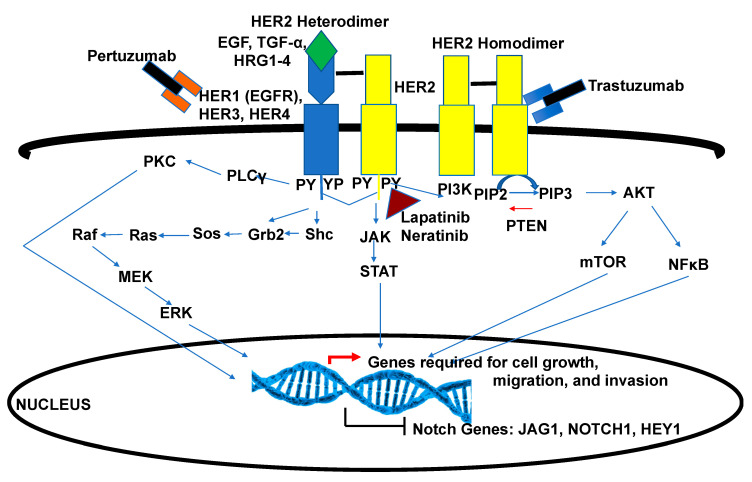
HER2 signaling and regulation of Notch. Gene amplification of the *ERBB2* gene results in overproduction of the HER2 protein. High expression of HER2 on the cell surface may promote homodimerization or heterodimerization with HER1, HER3, or HER4. Dimerization results in activation of the HER2 kinase domain and phosphorylation (P) of specific tyrosine (Y) residues within the cytoplasmic tail. The PY residues are docking sites for adaptor proteins including Grb2, Shc, the p85 subunit of PI-3Kinase, and others. HER2 signaling activates many kinase cascades, including the Ras-Raf-MEK-ERK, PKC, JAK-STAT, PI3Kinase-AKT-mTOR, and NF-κB pathways. Through a diverse transcriptional program, genes are expressed that promote cell-cycle progression, migration, and invasion. Other genes, including *NOTCH1*, *JAG1*, and *HEY1*, are repressed. Upon treatment with HER2-targeted therapy, using humanized monoclonal antibodies such as trastuzumab and pertuzumab or a small molecule tyrosine kinase inhibitor (lapatinib or neratinib), Notch genes are de-repressed. High Notch signaling promotes CSC enrichment, resistance, and increased tumor-initiating potential.

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
