# Peer review of "Notch Signaling in Breast Cancer: A Role in Drug Resistance"

_cells, 2020, doi:10.3390/cells9102204_

Round 1
Reviewer 1 Report
I general this is a very useful review of the role of Notch signaling in breast cancer. I do have some minor suggestions for additional references and comments which I hope the authors can incorporate into their manuscript.
There is an extensive discussion of the possibility of different BCSC types, ones that can be detected as CD44+CD24 - or as ALDH+.It has become clear that this different stem cell types differ in the extent to which the cells have undergone EMT; CD44+ is indicative of more mesenchymal types whereas ALDH+ is more epithelial, and perhaps more properly should be thought of as hybrid E/M. Exactly how these different states are related to the Notch signaling pathway and its coupling to EMT is an area of current invesgtigation and perhaps this could be highlighted. In this regard, the authors might consider citing
- Jolly, M.K.; Celià-Terrassa, T. Dynamics of Phenotypic Heterogeneity Associated with EMT and Stemness during Cancer Progression. J. Clin. Med. 2019, 8, 1542.
- Bocci, Federico, et al. "Toward understanding cancer stem cell heterogeneity in the tumor microenvironment." Proceedings of the National Academy of Sciences 116.1 (2019): 148-157
and citations therein to ongoing work of the Wicha group.
I don't agree with the statement that the cancer stem cell hypothesis must assume that the cancer initiating cell was stemlike. Again in the context of EMT (the original work by Mani and Weinberg) or in the context of drug resistance it has been observed that differentiated tumor cells can dedifferentiate to more stem like phenotypes. There is no reason on the context of this paper to "take sides" on this issue.
A more recent reference should be cited on the role of Jagged on IBC; this is contained in Jolly, M. K., Boareto, M., Debeb, B. G., Aceto, N., Farach-Carson, M. C., Woodward, W. A., & Levine, H. (2017). Inflammatory breast cancer: a model for investigating cluster-based dissemination. NPJ Breast Cancer, 3(1), 1-8.
In the discussion of the role of Notch signaling in normal breast tissue, there is cited the result that Notch signaling appears to be need for myoepithelial differentiation. As far as I recall, that paper did not track down whether the cells that become myoepithelial cells were high Delta or high Notch, under the usual assumption that Notch in the developmental context creates alternating patterns of different phenotypes. I think there is a chance of misunderstanding this issue in the current draft as most of the emphasis is on cells that themselves have high Notch (these are the ones that have stemless, resistance etc.) and I think it is quite likely that the myoepithelial cells are the high Delta ones. If the authors agree with my recollections, perhaps they could consider more carefully wording this brief discussion.
Author Response
Please see attached Word document for response to reviewer 1 comments.

Reviewer 2 Report
This is a very comprehensive review regarding an important area of breast cancer. The review was well organized and provided significant information and insight regarding Notch signaling in breast cancer. I only have some minor concerns over the editing of the MS. For example, on page 11, the first paragraph,
“Other studies demonstrated that HER2 stimulates Notch1 activation (109) thus possibly creating a positive feedback loop. Other studies demonstrated that HER2 blockade by trastuzumab or a tyrosine kinase inhibitor increases NOTCH1 and JAG1 expression activating Notch1 signaling to contribute to resistance (110). Specifically, trastuzumab resistant cells have increased expression of Notch1, JAG1, and its targets, including Hey1, Deltex1, and Hes5, and downregulation of Notch1 via siRNA sensitized these cells to trastuzumab treatment (Figure 5) (110).”
It is very confusing, especially regarding Ref. 110, if trastuzumab increases NOTCH1 and JAG1 expression, how trastuzumab resistant cells also has increased expression of Notch1 and JAG1?
Some English editing is also needed. For example, in the above text, two “Other studies” need to be edited. Also, in a sentence in the same paragraph, “Mechanistic studies showed resistance could be due direct regulation of the ERBB2 364 gene by Notch1 (108).” The author should add “that” following “showed” and add “to” after “due”.
The entire MS should be carefully edited.
Author Response
Please see attached cover letter and responses to reviewer 2.
